# Temporal Misinformation Detection: Simple Ways to Improve Temporal Generalization and Better Evaluate Language Models

## Abstract

Most of the current misinformation detectors display a lack of temporal generalization, despite the increasing reported scores in the literature. This phenomenon can be attributed to classical machine learning evaluation protocols based on random splits. While widely adopted, these protocols often fail to reflect real-world model performance, a limitation that is particularly critical in misinformation detection, where temporal dynamics play a central role. In this paper, we present a comprehensive analysis of temporal biases across multiple misinformation datasets, with a specific focus on the temporal distribution of labels. We also introduce simple yet effective methods to improve performance in scenarios where temporal generalization is critical for NLP tasks. Our findings show that classical evaluation protocols tend to overestimate model performance in misinformation detection. To address this, we propose FC30, a new dataset, and introduce a general-purpose evaluation metric designed to better assess models under temporal shift and capture potential temporal bias.

## 1 Introduction

Misinformation detection and related sub-tasks, such as fake news detection, rumor detection, or stance classification, are of utmost importance in the current geopolitical context. Misinformation has adverse effects on society, ranging from influencing elections to creating a climate of defiance of the population against institutions.

To counter misinformation, researchers and industrials have proposed solutions based on annotated data from websites such as Snopes[1] or PolitiFact[2] to create evaluation datasets with human annotations. These evaluation datasets are then used, following rigorous machine learning protocols, to measure machine learning systems performance in misinformation detection.

However, the standard NLP practice of randomly splitting data into training, validation, and test sets may not be well-suited for misinformation detection. This task is particularly prone to knowledge leakage, as articles covering the same or related events can appear across splits, leading models to learn event-specific representations rather than generalizable misinformation patterns Wang et al. (2018). Moreover, some datasets are relatively old, increasing the likelihood that test articles were seen during the language model pre-training. In practice, a robust misinformation detection system should perform well not only in articles published after the last article used for training, but also in those released after the model's pre-training period.

In this paper, we propose to study how accounting for the temporality of misinformation datasets can help better evaluate misinformation detection models and learn text representations that remain relevant over time. By temporality, we refer to the chronological ordering of data based on their timestamps, which may help capture evolving trends in future data. While this is a common concern for time series and temporal forecasting, the complexity of learning both text representations and their temporal dynamics makes traditional time series models hardly usable for NLP tasks. Our research questions and contributions are the following:

---

[1]https://www.snopes.com/
[2]https://www.politifact.com/

- **Q1. Do current misinformation benchmarks correctly evaluate model performance in practice?** We show, through experiments on a wide range of misinformation datasets, that classical evaluation methods based on random splits tend to overestimate encoder models when compared to models trained and evaluated with a temporal test set.

- **Q2. How to define a more robust framework for measuring temporal generalization? 1. Metric for temporal bias in labels**. e propose a general data-agnostic metric, named LabDrift to quantify label drift in datasets. This metric differs from standard statistical tests such as Cramér's V, focusing more on the theoretical performance of models if labels were biased. **2. New extended fact-checking dataset**. We collect and share FC30, a new dataset based on claims covering a time period of 30 years, with a large proportion of claims made after the pre-training of popular language models based on encoders. **3. Approaches for improving temporal generalization**. We propose two new baselines, named *Fusion* and *TempoGen*, to improve temporal generalization and show performance gains in a temporal setting on several models, including models published before and after the test set period. These approaches guide what the learned features represent by adequately choosing the validation split, leading to better-lasting text representations.

## 2 RELATED WORK

Misinformation detection is usually performed through several tasks, such as detecting fake news or rumors. Many datasets exist (D'Ulizia et al., 2021), allowing for the training of models to detect different aspects of misinformation. Fake news detection is a popular task that is often resolved with approaches that check the consistency between several articles (Fung et al., 2021), or by using external knowledge in the model (Dun et al., 2021). Another related task is rumor detection, less based on knowledge and more on stylistic (Shu et al., 2019) and conversational (Zubiaga et al., 2015) features. Misinformation is also linked to bad practices in journalism, making the task of *clickbait* detection (Zheng et al., 2018) also of interest. On social media, the stance of comments responding to a main post helps detect fake news (Riedel et al., 2018), allowing for leveraging crowd answers for automatic fake news detection (Taranukhin et al., 2024).

In this paper, we are interested in generalizing to future data (model performance when applied to data that was not seen during training and that comes from a later time period), also sometimes called temporal alignment (Luu et al., 2022), which is similar to the task of domain adaptation. However, the existing literature on temporality and domain adaptation (Chen et al., 2020; Song et al., 2021; Ragab et al., 2023) is either not transferable to textual tasks or too specific to the characteristics of the computer vision tasks considered. For misinformation detection, EANN (Wang et al., 2018) has been proposed to help generalization to new topics, but did not evaluate the temporal context evolution between the topics, or the temporality of each topic, which may vary. Improving over EANN, Forecasting Temporal Trends (Hu et al., 2023) attempts to incorporate temporal forecasting of embeddings to reweight the importance of each sample before training a language model. A similar approach for graph-based misinformation detection has been proposed (Kim et al., 2025). Other works on improving generalization exist, but they primarily address the issue of catastrophic forgetting - where models forget previously learned information when trained on new data (Wang et al., 2025) - or focus on generalization across different datasets (Qin & Zhang, 2024). For help in generalization, three axes were identified by Verhoeven et al. (2025): time, content, and publisher. However, their temporal analysis is limited due to an imbalance in the label distribution.

More focused on temporality, Su et al. (2022) shows that language models perform worse over time due to lexical semantic change, and that domain adaptation methods only aggravate temporal misalignment. An extended analysis has been performed that confirms temporal misalignment in BERT (Röttger & Pierrehumbert, 2021), with degradation in performance when the test set is far from the training set. The issue of temporal misalignment is also relevant for LLMs, with several identified biases (Kishore & He, 2024; Zhu et al., 2025). GenBench (Stepanova & Ross, 2023) is a benchmark designed to analyze the performance of multimodal misinformation detectors over time on a single misinformation dataset.

In this paper, we focus on the generalization of learned features on future data, in the sense of unseen data published after training, as in a real application case. The new data covers different topics, new words, and new entities, and comes from different sources, making the use of time series

or distribution shifts methods irrelevant, as no trends can be defined. We propose a new rigorous formalization of this problem to design more robust benchmarks for misinformation detection, better reflecting the performance obtained in applicative cases, where the evolution of data can not be predicted.

## 3 TEMPORAL LABEL BIAS IN MISINFORMATION DATASETS

In this section, we propose and formalize the constraints that a dataset must satisfy to properly evaluate temporal generalization, present analyses on the impact of such a setting across multiple misinformation datasets, and introduce a metric to measure temporal label bias distribution, also called label drift.

### 3.1 TEMPORAL GENERALIZATION: ASSUMPTIONS, CONSTRAINTS, AND DEFINITIONS

Before going further, we present our definition of temporal generalization. In a classical machine learning setting, the data is randomly distributed among the training, validation, and testing splits. The trained model is chosen based on the best score on the validation set, then the score on the test set is considered the expected performance of the model when deployed, as the test set is composed of unseen data during training. However, for tasks such as misinformation detection, we propose adopting a temporal evaluation setting, similar to those used in time series analysis, where models are trained on past data and evaluated on future instances. This setup better reflects real-world deployment conditions and helps prevent information leakage over time.

Let us suppose that available data, with $N$ samples (noted $(X_k, y_k), k \in [\![1, N]\!]$, with $X$ the input of the model and $y$ the corresponding label), is provided with associated timestamps $t_k$. Data samples can then be **T**emporally **O**rdered at the time of splitting the dataset (Constraint TO).

$$\forall (i,j) \in [\![1, N]\!]^2,\ i < j \Rightarrow t_i \leq t_j \tag{TO}$$

We argue that for some NLP tasks, such as misinformation detection, a temporal split should be considered for the test split, by applying a **T**est **C**ondition, as written in Constraint TC, where $T_{train}, T_{valid}$ and $T_{test}$ are the lists of timestamps of data used in training, validation and test splits, respectively. This means that knowledge leakage is not possible from training and validation to the test split.

$$\max\left(\max\left(T_{train}\right), \max\left(T_{valid}\right)\right) < \min\left(T_{test}\right) \tag{TC}$$

In addition to these constraints, we propose an additional requirement related to the **P**re-**T**raining date of the pre-trained models, namely $t_{pre-training}$ stated in Constraint PT . This would allow for a more realistic setting, as it guarantees that the test data was never seen during the pre-training of the model (this is especially important for NLP tasks).

$$t_{pre-training} < \min\left(T_{test}\right) \tag{PT}$$

These last two constraints TC and PT account for several concerns we faced in misinformation detection that could also benefit other classification tasks - generalization to new topics, evolution of language, and knowledge leakage from random splits.

We consider that a model evaluated in a setting that satisfies the three constraints allows us to measure **temporal generalization**, as all data in the test set is unseen by the model in terms of knowledge, context, and language.

While this makes sense in a temporal series or sequential learning context, it has not been explored in NLP yet, where the complexity of learning text representations has obfuscated the need for temporal generalization, which is not necessary for most NLP tasks.

Table 1: *Meta-performance* of models in a classic (noted C) and temporal (noted T) evaluation protocol. Difference significance is computed with a t-test and we report p-values (*$p < 0.05$, **$p < 0.01$, ***$p < 0.001$).

| Dataset | Classical evaluation | | Realistic temporal evaluation | |
|---|---|---|---|---|
| | C - Acc. | C - F1 | T - Acc. | T - F1 |
| GossipCop | **75.04**±3.80 | **74.48**±3.73 | 70.09* | 69.94* |
| ISOT | **87.05**±0.18 | **87.01**±0.18 | 86.72** | 79.21*** |
| MediaEval | **86.14**±3.67 | **86.01**±3.70 | 54.13*** | 43.09*** |
| MisInfoText | 67.02±7.69 | 45.02±11.16 | **77.34*** | **50.82** |
| PHEME | **80.66**±9.29 | **51.14**±15.32 | 56.67*** | 36.17 |
| PolitiFact | **93.33**±5.03 | **93.14**±5.33 | 92.31 | 67.95*** |
| Proppy | **97.82**±0.47 | **94.46**±1.05 | 51.66*** | 45.61*** |
| FC30 | 72.3±0.78 | **65.44**±0.36 | **78.08***  | 51.22*** |

## 3.2 THE MEASURED PERFORMANCE IS OVER-ESTIMATED

Because of the risk of temporal bias for misinformation detection, we aim at measuring whether the perceived performance of models (which is protocol-dependent, called *meta-performance*) in a classical setting and a temporal generalization setting align. To do so, we performed two sets of experiments on various misinformation detection datasets, including fake news detection, fact-checking, and stance detection:

- A multiclass classification (binary for all datasets except MisInfoText and FC30, which are ternary) using `deberta-v3-large`. The splits (including test) are randomly generated. These experiments with classic splits are noted **C**.

- The same multiclass classification problem, but with splits verifying Constraints TO and TC. The test set consists of the last 10% of the published data, while the remaining data is randomly split 80/10 for training and validation. Experiments using this split are labeled **T** for temporal.

We report the accuracy and F1-score on the test set averaged over five runs in Table 1. The numbers reported are not to be considered as standard performance metrics, but as perceived *meta-performance*. They only reflect the protocol-dependent metrics in the experimental setting, which is the expected performance of the model under the protocol assumptions. Random splits assume that features are equally shared independent of any other factor, while the temporal generalization (also called temporality-aware in a graphical context by Kim et al. (2025)) assumes that the real performance of model is measured on the last published data, which is the case for misinformation detectors. Reported numbers in Table 1 can not be compared for model selection.

We observe that models are significantly over-evaluated in the classic setting, with an average increase of 11.54 accuracy points and 19.08 F1-score points compared to the temporal setting. This can be explained by possible knowledge leakage that occurs with random split creation, which does not correspond to real use cases. Furthermore, the significant drop in *meta-performance* under the temporal setting suggests that models trained without temporal constraints are less effective at handling future misinformation, limiting their practical utility.

An additional challenge arises with smaller datasets like PolitiFact (only 38 samples in test set), where the temporal test set may contain very few samples for certain classes —for example, only one instance of real news—leading to highly biased performance metrics. This issue is mitigated in larger datasets, with more samples for each label.

In addition to previous observations, there could also be an issue with temporal splits if labels are not evenly distributed over time. For fake news detection, this could lead to models over-trained on fake news of a specific time frame, making learned features not robust for temporal generalization. To avoid this possible issue, we propose a metric for measuring temporal biases in dataset labels in the next section.

### 3.3 LABDRIFT, A NEW MEASURE FOR LABEL DRIFT

Our proposition for measuring whether a dataset is temporally biased is to create a metric valued at 1 if the temporally ordered labels are totally separated and close to 0 if the labels are evenly distributed over time. This metric measures the concept of label shift proposed by Yee (2025), though it lacks practical methods for measurement.

Let a list of $N$ temporally ordered labels $Y^{data} = (y_i)_{i \in [\![1,N]\!]}$ with associated timestamps verifying Constraint TO. If the labels were totally biased, all 'legitimate' labels would be grouped first, then all 'fake' labels (or the opposite, depending on the dataset). We note this hypothetical biased list of labels $Y^{biased}$. More precisely, $Y^{data}$ and $Y^{biased}$ contain the same labels, but are temporally ordered for $Y^{data}$, and grouped by labels for $Y^{biased}$. The metric measures how well a model can predict based only on the publication date, assigning one label before a certain date and another label after that date. The list of predicted labels by a model predicting the first class $i$ times, then the other $(N - i)$ times, is named $Y_i$. We can plot the performance of this classifier by visualizing its F1-score with varying $i$, following the piecewise constant functions defined in Equations 1 and 2. These functions represent the date-biased predictor F1-score depending on the threshold date used for predictions. $f^{bias}$ is the performance to predict $Y^{biased}$, and $f^{data}$ the performance to predict $Y^{data}$.

$$f^{bias}\left(\frac{i-1}{N-2}\right) = \text{F1}\left(Y_i, Y^{biased}\right), \forall i \in [\![1, N-1]\!] \tag{1}$$

$$f^{data}\left(\frac{i-1}{N-2}\right) = \text{F1}\left(Y_i, Y^{data}\right), \forall i \in [\![1, N-1]\!] \tag{2}$$

The two curves are plotted and cover an area. The metric, LabDrift, noted $\lambda$, is the ratio of the (absolute) area covered by $f^{data}$ and the area covered by $f^{bias}$, as defined in Equation 3.

$$\lambda = \frac{\int_{[0,1]} \left| f^{data}(x) - \frac{f^{data}(0) + f^{data}(1)}{2} \right| dx}{\int_{[0,1]} \left| f^{bias}(x) - \frac{f^{bias}(0) + f^{bias}(1)}{2} \right| dx} \tag{3}$$

An example is given in Figure 1 on the PolitiFact dataset, with $f^{bias}$ plotted in orange and $f^{data}$ plotted in blue. This dataset has highly temporally biased labels, with performance reaching more than 89% of F1-score with a threshold date at 46% of the whole temporally ordered dataset, which is confirmed with the high bias metrics ($\lambda = 0.9310$).

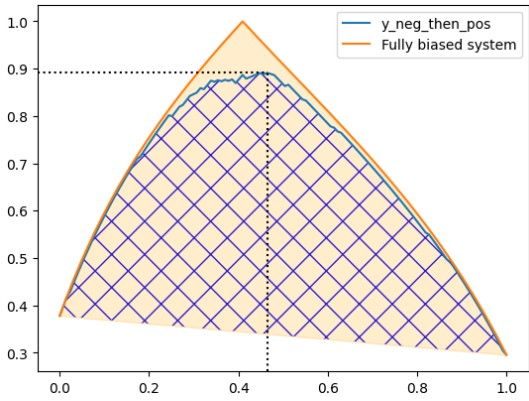

Figure 1: Visual explanation of the temporal bias metric on the PolitiFact dataset ($\lambda = 0.9310$, ratio of the area hashed in blue over the orange area). The maximal F1-score achievable by a system classifying only based on publication date can be read on the blue curve (see dotted lines).

This computation is for binary classification. When the problem is multiclass, the metric has to be computed for each possible pair of classes, and these partial biases are averaged using a harmonic

mean, as partial biases are ratios. The advantage of such a metric is that it is data-agnostic and can be computed using just a list of temporally ordered labels.

We have performed an analysis on many published misinformation datasets and reported the bias metrics when the dataset and temporal information were available. Results are given in Table 2. Information about all considered datasets for the analysis are given in Appendix A. About half of the datasets provide temporal information, and the measured temporal biases vary highly.

Table 2: Review on temporal bias in labels in misinformation datasets, with the $\lambda$ metric when it is possible to compute. The closer to 0 the metric is, the less biased the labels of the dataset.

| Name | LabDrift $\lambda$ |
|---|---|
| Buzzface (Santia & Williams, 2018) | 0.3123 |
| CREDBANK (Mitra & Gilbert, 2021) | 0.8788 |
| GossipCop (Shu et al., 2019) | 0.5198 |
| MediaEval (Boididou et al., 2017) | 0.1514 |
| MisInfoText (Asr & Taboada, 2019) | 0.3958 |
| PHEME (Zubiaga et al., 2015) | 0.4236 |
| PolitiFact (Shu et al., 2019) | 0.9310 |
| Proppy (Barrón-Cedeño et al., 2019) | 0.7961 |
| FC30 (Ours) | 0.1130 |

## 4 FC30: AN EXTENDED DATASET FOR TEMPORAL BIAS EXPERIMENTS

To the best of our knowledge, there is no existing dataset that allows a setting verifying Constraints TO, TC, and PT for the use of popular NLP encoders such as BERT (Devlin et al., 2019), and DeBERTaV3 (He et al., 2021) (respectively trained in 2018 and 2020). To help bias analysis in a temporal setting, we propose FC30, a new fact-checking dataset that allows verification of the three constraints to measure temporal generalization with adequate data splits satisfying the previously defined constraints (see Section 3.1).

FC30 is based on collected fact-checks from Snopes and PolitiFact, covering a range of 30 years, from 1995 to 2025. It is composed of 36,619 fact-checks with 25 label types, which can be grouped into three main labels. More details are provided in Appendix B.

The dataset has several advantages in responding to the limits identified in the literature:

- The time range of the data is large, allowing for evaluation on different time frames, and the proportion of true, false, and mixed labels is relatively constant over time.

- A large amount of data was published after the training of the most popular NLP encoder models, with 30% of claims published after 2020, the training cut-off date of DeBERTaV3, allowing to verify Constraint PT for the most popular NLP encoders - except for Modern-BERT (Warner et al., 2024) and EuroBERT (Boizard et al., 2025).

- Thanks to the time range, and internationality of topics analyzed by Snopes and Politi-Fact, new topics and vocabulary appear over time, allowing to perform experiments on new topics (Covid-19 crisis hit in 2020, and major political events have taken place since 2020).

The dataset with the main labels is available at GITHUB_LINK (in supplementary materials).

## 5 ENHANCING TEMPORAL GENERALIZATION

In this section, we propose *Fusion* and *TempoGen*, two approaches that have the advantage of being both simple and effective to improve temporal generalization as defined in Section 3.1. The proposed approaches significantly improve performance in a temporal context, reaching levels similar to what is measured in a classical setting for some models.

## 5.1 EXPERIMENTAL SETUP

For the remainder of the paper, we are in a temporal generalization scenario, verifying Constraints TO and TC, with the last 10% of data left out for testing. We define two other splits for the training phase:

- Random split: The remaining 90% of data are randomly split into training and validation. This is supposed to favor temporally invariant features, which may not be the best choice, as demonstrated in Section 3.2. Features learned with this split are expected to be time-invariant, as they appear in all periods of time in the dataset.

- Temporal split: The remaining 90% of data are temporally split, verifying Equation 4. This split is supposed to favor temporal features that better align with future data.

$$\max\left(T_{train}\right) < \min\left(T_{valid}\right) \tag{4}$$

All the following experiments are conducted as multiclass text classification on FC30, with reported performance on the temporal test set. We chose to work with encoder models, as they have shown to be proficient baselines (Pelrine et al., 2021). Experiments were conducted using three families of models, BERT (Devlin et al., 2019), DeBERTaV3 (He et al., 2021), and EuroBERT (Boizard et al., 2025). We chose two variants for each family of models (base and large or equivalent). These models were chosen as BERT in this setting verifies Constraint PT, DeBERTaV3 has been the state-of-the-art encoder for several years, and EuroBERT was published after the last data of FC30.

Training was performed with the transformers `Trainer` class, with the model reaching the highest F1 score on validation being restored before evaluation on the temporal test set. The models trained on the random and temporal split are named **Random** and **Temporal**, respectively (noted R and T in tables).

## 5.2 FUSION

After the training on these two splits, we propose to compute all embeddings obtained by the **Random** and **Temporal** models, and to train a two-layer perceptron model on a concatenation of both embeddings with a temporal split. This model is called **Fusion** (noted F in tables) and can leverage all random and temporal information, but require both inferences from **Random** and **Temporal**.

In addition to this model, we propose in the next section **TempoGen**, an approach to improve generalization with no additional costs at inference.

## 5.3 TEMPOGEN

To describe the intuition behind TempoGen, we have to display the learned embeddings by the **Random** and **Temporal** models (examples are given using DeBERTa large). To do so, all embeddings are projected onto the principal components of the features obtained with the **Random** model and visualized on the same plot (see Figure 2 for embeddings distribution in 3D and corresponding decision frontier in 2D).

We observe that the random embeddings (on the left) are clearly separated by class. The temporal embeddings are closer, but have a different decision frontier. Working with both types of embeddings (two data samples per article, one from the random split and one from the temporal split), the decision frontier is a mix between both, conserving temporal and general decision frontiers where it matters.

The main idea of **TempoGen** (noted TG in tables) is to use both Random and Temporal embeddings during training to obtain a modified decision frontier that would fit the two distributions. When training TempoGen, the Random and Temporal models have to be trained first, doubling the cost of training. After that, a small two-layer perceptron is trained with both types of embeddings in the dataset (two samples for each article, one with random embeddings, one with temporal embeddings, both labeled the same), with minimal additional cost compared to the training of the encoder models. However, for inference, only the temporal embeddings are used, as they are better aligned with data to come in the near future, leading to no additional costs after the TempoGen model training.

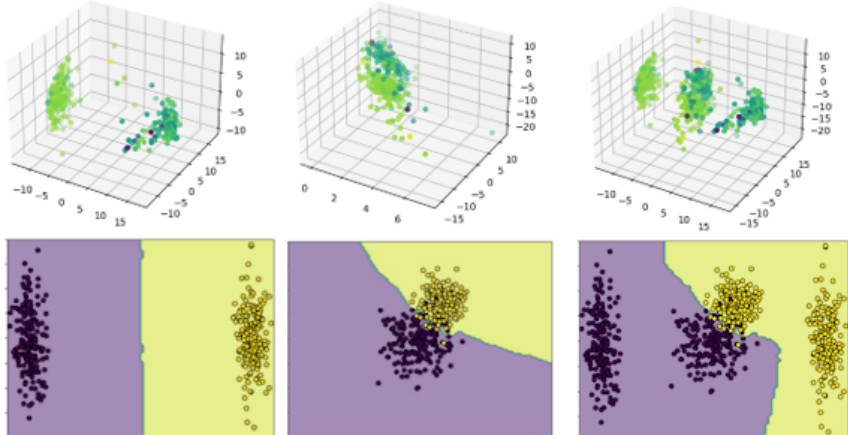

Figure 2: PolitiFact embeddings distribution on the top and associated decision frontiers on the bottom. From left to right, the **Random** embeddings, the **Temporal** embeddings, and both of them on the same plot. Fake news labels are in a darker color, and legitimate news labels are lighter.

## 5.4 RESULTS

The main results are shown in Table 3. Because the number of elements in each class is unbalanced, the primary metric to prioritize is the F1 score. We report the mean rank of each approach on the six tested models.

Table 3: Temporal generalization of language models in different training scenarios. Best accuracies and F1 are in bold, and second-best results are highlighted by the $\dagger$ symbol.

| Model | R Acc | R F1 | T Acc | T F1 | F Acc | F F1 | TG Acc | TG F1 |
|---|---|---|---|---|---|---|---|---|
| BERT base | **78.63** | 51.56 | 75.77 | 47.95 | 78.27$^\dagger$ | **54.70** | 78.10 | 54.53$^\dagger$ |
| BERT large | **78.26** | 50.21 | 76.34 | 48.79 | 77.54$^\dagger$ | **55.28** | 76.30 | 52.00$^\dagger$ |
| DeBERTa base | **79.67** | 52.50 | 78.23$^\dagger$ | 51.30 | 77.49 | **55.00** | 75.30 | 53.33$^\dagger$ |
| DeBERTa large | 78.08$^\dagger$ | 51.22 | **80.05** | 52.99 | 77.46 | 58.99$^\dagger$ | 77.84 | **60.25** |
| EuroBERT-210M | **80.38** | 54.08$^\dagger$ | 78.41 | 51.61 | 79.69$^\dagger$ | **59.79** | 74.53 | 45.77 |
| EuroBERT-610M | 78.61$^\dagger$ | 51.51$^\dagger$ | **79.27** | **52.28** | 76.86 | 50.87 | 74.36 | 44.60 |
| Mean ranking | **1.33** | 2.83 | 2.33$^\dagger$ | 3.16 | 2.66 | **1.5** | 3.66 | 2.5$^\dagger$ |

Several observations can be made. Globally, the use of the temporal split alone does not significantly improve performance, which is lower than the random split in most cases. However, combining both representation types in the Fusion model is the best approach, achieving the best score in two-thirds of the experiments. TempoGen is the second-best approach, but has a lower inference cost, half of the Fusion approach. For models verifying Constraint PT (BERT and DeBERTa), TempoGen allows to reach performance similar to more recent models not verifying Constraint PT, for which the impact of the Fusion and TempoGen approaches is less significant. Even with TempoGen and the Fusion approach, the measured performance with DeBERTa large is still lower than what was measured in a classical machine learning setting, confirming our observations that traditional machine learning evaluation over-evaluates misinformation detection models. To further support these claims, more experiments were conducted on other temporal misinformation datasets, with results reported in Appendix C, leading to the same conclusions on most datasets. Some tasks (such as propaganda or rumor detection) are less affected by temporal evolution of features, making representations learned on classical split relevant even in the future.

To better understand how the proposed approaches affect performance, we show in Figure 3 the performance DeBERTa large on five-year time windows of FC30, covering training data first, validation, and test. The other models have similar curves, given in Appendix D.

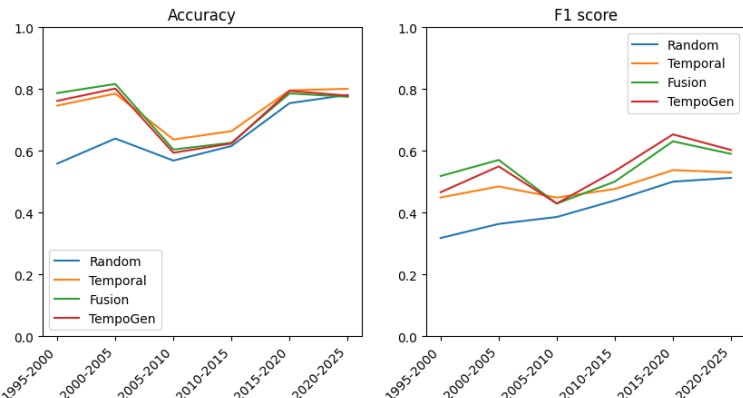

Figure 3: DeBERTa large performance over time for different training scenarios.

For models verifying Constraint PT (BERT and DeBERTaV3), the F1-scores display an *m-shaped* curve. All models have difficulties in the 2005-2010 time window. The high performance observed on non-random approaches between 2015 and 2020 is due to model selection based on temporal split, leading to a slight loss in performance on the test set from 2020 to 2025. However, this drop in performance usually reaches a higher performance than a random split, showing the interest of the proposed approaches in improving temporal generalization. For EuroBERT, which has seen relevant context during its pre-training, results are mixed, but do not reflect real use cases with test data published after the model pre-training.

In the general case, combining temporally-selected and classical embeddings in the Fusion approach improves temporal generalization on new data, but at twice the cost of inference and training. If speed is more important, the TempoGen approach also improves temporal generalization, with twice the cost of training, but no additional cost at inference.

## 6 CONCLUSION AND FUTURE WORK

In this paper, we have introduced a temporal generalization formalization, allowing for a more realistic evaluation of misinformation detection systems. To evaluate models in this configuration, we proposed FC30, a new extended dataset containing claims covering 30 years of fact-checking, with a significant part published after the pre-training of popular encoder models. To further support the measurement of temporal biases, we proposed LabDrift, a data-agnostic metric to assess label drift, which could hinder training. We then showed that classical machine learning evaluation over-evaluates misinformation detection systems, based on experiments on eight datasets, and proposed efficient yet straightforward approaches to improve temporal generalization.

Several challenges remain to be addressed in future work:

- Generally, performance improvements in NLP on benchmarks are difficult to impute. They may be due to architecture improvements, but also to knowledge leakage due to the addition of data in the pre-training of models. These two factors are usually modified at the same time in newly proposed models, making it hard to determine if the architecture is making a difference or if it is an unexpected effect of adding pre-training data. We hope this work will help design benchmarks for *temporality-sensitive* tasks, enabling better evaluation and comparison of models.

- Extending the formalization to external knowledge-enhanced tasks, with additional constraints on retrieved information to guarantee reproducible results, even after the evolution of the external knowledge base.

- Extending the use of the proposed *Fusion* and *TempoGen* approaches to other tasks solved by encoders that could benefit from temporality in datasets, such as computer vision (Pégeot et al., 2025).

## ETHICS STATEMENT

This paper deals with the topic of misinformation in a practical context, proposing a robust protocol for better evaluating misinformation detection systems. This topic is of critical interest for society, as misinformation has a large impact on individuals and society as a whole.

The proposed benchmark and data can be used to train misinformation detection systems, but contains by nature harmful content for misinformation examples. We do not give examples in the paper, but the repository can contain offensive or harmful content.

## REPRODUCIBILITY STATEMENT

To ensure the reproducibility of the presented results, we would like to share all components that led to the production of the reported findings in the following Github GITHUB_LINK (in supplementary materials). This repository contains the following elements:

- **Data**: for produced datasets (FC30), data is provided with loading functions. More details on the dataset composition is also given in Appendix B. Other datasets have their own license so we do not share them. They are however correctly cited in the paper, and are accessible online at the time of submission.
- **Metrics**: the LabDrift metric code is provided in its own Python file, as well with examples in an associated Jupyter notebook

## ON THE USE OF LARGE LANGUAGE MODELS

In the writing of this paper, AI-based tools (not based on LLMs) were used to improve readability, clarity and quality of language. All scientific content has been produced without the help of any AI-based model, including problem formulation, experiment design, metric and code production, as well as interpretations of results and analyses.

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

## A  TEMPORAL BIASES IN LABEL DISTRIBUTION OF EXISTING DATASETS

We give the detail of all considered dataset for temporal biases in label distribution in Table 4. It is the detailed version of Table 2.

Table 4: Review on temporal bias in labels in misinformation datasets, with the $\lambda$ metric when it is possible to compute.

| Name | Data available | Timestamps | LabDrift $\lambda$ |
|---|---|---|---|
| Burfoot (Burfoot & Baldwin, 2009) | Dead link | NA | - |
| Buzzface (Santia & Williams, 2018) | Yes (behind API) | Yes | 0.3123 |
| CREDBANK (Mitra & Gilbert, 2021) | Yes | Yes | 0.8788 |
| EMERGENT (Ferreira & Vlachos, 2016) | Yes | No | - |
| FacebookHoax (Tacchini et al., 2017) | Yes (behind API) | Yes | - |
| FCV-2018 (Papadopoulou et al., 2019) | Yes | No | - |
| FEVER (Thorne et al., 2018) | Yes | No | - |
| FNC-1 (Riedel et al., 2018) | Yes | No | - |
| GossipCop (Shu et al., 2019) | Yes | Yes | 0.5198 |
| Horne2017 (Horne & Adali, 2017) | Yes | No | - |
| LIAR (Wang, 2017) | No | - | - |
| MediaEval (Boididou et al., 2017) | Yes | Yes | 0.1514 |
| MisInfoText (Asr & Taboada, 2019) | Yes | Yes | 0.3958 |
| NELA-GT-2018 (Nørregaard et al., 2019) | Deaccessioned | NA | - |
| Ott et al. (Ott et al., 2011) | Dead link | - | - |
| PHEME (Zubiaga et al., 2015) | Yes | Yes | 0.4236 |
| PolitiFact (Shu et al., 2019) | Yes | Yes | 0.9310 |
| Proppy (Barrón-Cedeño et al., 2019) | Yes | Yes | 0.7961 |
| Spanish fake news (Posadas-Durán et al., 2019) | Yes | No | - |
| Tam et al. (Tam et al., 2019) | No | NA | - |
| TSHP-17 (Rashkin et al., 2017) | Yes | Yes (one class) | - |
| TW_info (Jang et al., 2019) | No | NA | - |
| Vlachos and Riedel dataset (Vlachos & Riedel, 2014) | No | NA | - |
| Yelp dataset (Barbado et al., 2019) | Yes | No | - |
| Zheng et al. (Zheng et al., 2018) | Yes | No | - |
| FC30 (Ours) | Yes | Yes | 0.1130 |

## B  THE FC30 DATASET

Data is composed of 36,619 claims collected from PolitiFact (https://www.politifact.com/) and Snopes (https://www.snopes.com/), covering a time period of approximatively 30 years from September 24, 1995 to March 4, 2025. Claims are a string that usually contain two information:

- The speaker or origin of the claim. It can be a political figure, or a social media without the name of the user.
- The content of the claim.

Here is an example of claim, the latest one contained in the dataset: "A photograph authentically showed Ukrainian President Volodymyr Zelenskyy and first lady Olena Zelenska posing in front of stacks of money.". The structure of the claim can change depending on the source (PolitiFact or Snopes) and time of publication (the fact-checking template has evolved over time).

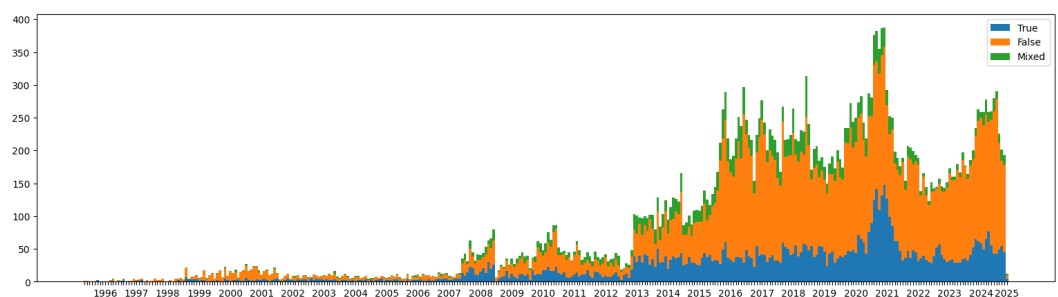

Figure 4: Main labels distribution over time in FC30.

Data is stored in the form of JSON file with the following schema:

```
{
    "claim": str,
    "label": str,
    "year": int,
    "month": int,
    "day": int,
}
```

The originally extracted labels are multiple, but can be grouped into three main categories. We provide below the dictionary we used to group the 25 fine labels into three larger categories for experiments.

```
fine_annotation_to_label = {
    'False': "false",
    'Half True': "mixed",
    'Mostly False': "false",
    'True': "true",
    'Mostly True': "true",
    'Pants on Fire': "false",
    '': "mixed",
    'Labeled Satire': "false",
    'Miscaptioned': "false",
    'Fake': "false",
    'Incorrect Attribution': "false",
    'Mixture': "mixed",
    'Unfounded': "false",
    'Correct Attribution': "true",
    'Scam': "false",
    'Research In Progress': "mixed",
    'Unproven': "false",
    'Originated as Satire': "false",
    'Recall': "mixed",
    'Outdated': "false",
    'Legend': "false",
    'Legit': "true",
    'Lost Legend': "false",
    "Mixed": "mixed",
    "No label": "mixed",
}
```

We show the label distribution over time in Figure 4.

## C  PERFORMANCE OF MODELS ON OTHER DATASETS

Table 5: Temporal generalization of language models in different training scenarios on the Gossip-Cop dataset. Best accuracies and F1 are in bold, and second-best results are highlighted by the $\dagger$ symbol.

| Model | R Acc | R F1 | T Acc | T F1 | F Acc | F F1 | TG Acc | TG F1 |
|---|---|---|---|---|---|---|---|---|
| BERT base | **77.78** | **77.70** | 67.52 | 67.40 | 70.94$^\dagger$ | 70.94$^\dagger$ | 70.09 | 70.09 |
| BERT large | 76.07 | 75.98 | 73.50 | 73.43 | **82.91** | **82.89** | 81.20$^\dagger$ | 81.16$^\dagger$ |
| DeBERTa base | 71.79 | 71.76 | 75.21 | 74.74 | 76.07$^\dagger$ | 76.05$^\dagger$ | **76.92** | **76.92** |
| DeBERTa large | **79.49** | **79.49** | 73.50 | 73.43 | 74.36 | 74.21 | 76.92$^\dagger$ | 76.81$^\dagger$ |
| EuroBERT-210M | 71.79$^\dagger$ | 71.59$^\dagger$ | 69.23 | 68.72 | **74.36** | **74.21** | 50.43 | 35.00 |
| EuroBERT-610M | **76.92** | **76.86** | 70.09 | 68.27 | 75.21$^\dagger$ | 74.74$^\dagger$ | 64.96 | 61.29 |
| Mean ranking | 2$^\dagger$ | 2.16$^\dagger$ | 3.5 | 3.5 | **1.83** | **1.83** | 2.66 | 2.66 |

Table 6: Temporal generalization of language models in different training scenarios on the ISOT dataset. Best accuracies and F1 are in bold, and second-best results are highlighted by the $\dagger$ symbol.

| Model | R Acc | R F1 | T Acc | T F1 | F Acc | F F1 | TG Acc | TG F1 |
|---|---|---|---|---|---|---|---|---|
| BERT base | **88.75** | **81.28** | 87.46$^\dagger$ | 80.13$^\dagger$ | 83.38 | 76.26 | 83.83 | 76.25 |
| BERT large | **87.57** | **79.97** | 85.94 | 78.37 | 85.34 | 78.38 | 86.26$^\dagger$ | 79.19$^\dagger$ |
| DeBERTa base | **89.00** | **81.45** | 87.66$^\dagger$ | 79.59$^\dagger$ | 86.08 | 78.54 | 84.70 | 76.84 |
| DeBERTa large | **88.71** | **81.33** | 86.75$^\dagger$ | 79.07 | 86.66 | 79.32$^\dagger$ | 83.38 | 75.64 |
| EuroBERT-210M | **88.39** | **81.29** | 88.04$^\dagger$ | 80.75$^\dagger$ | 85.41 | 78.24 | 82.16 | 75.08 |
| EuroBERT-610M | **87.77** | **80.51** | 84.74 | 76.43 | 86.43$^\dagger$ | 79.29$^\dagger$ | 81.96 | 74.96 |
| Mean ranking | **1** | **1** | 2.33$^\dagger$ | 2.66$^\dagger$ | 3.16 | 2.66$^\dagger$ | 3.5 | 3.66 |

Table 7: Temporal generalization of language models in different training scenarios on the MediaEval dataset. Best accuracies and F1 are in bold, and second-best results are highlighted by the $\dagger$ symbol.

| Model | R Acc | R F1 | T Acc | T F1 | F Acc | F F1 | TG Acc | TG F1 |
|---|---|---|---|---|---|---|---|---|
| BERT base | 68.81$^\dagger$ | **53.89** | 59.17 | 45.57 | 67.43 | 50.70$^\dagger$ | **71.56** | 50.60 |
| BERT large | 53.67 | 42.81 | 67.43$^\dagger$ | 47.17 | **71.56** | **52.54** | 67.43$^\dagger$ | 48.12$^\dagger$ |
| DeBERTa base | 36.70 | 33.75 | 66.51$^\dagger$ | 43.57 | 57.80 | 45.97$^\dagger$ | **90.83** | **47.60** |
| DeBERTa large | 48.62 | 42.72 | 22.02 | 21.85 | **59.63** | **48.88** | 59.63 | 45.16$^\dagger$ |
| EuroBERT-210M | 66.51 | 52.30$^\dagger$ | 55.96 | 43.62 | 71.10$^\dagger$ | **53.97** | 72.48 | 52.19 |
| EuroBERT-610M | 69.27 | **55.58** | 81.65$^\dagger$ | 51.39 | 65.60 | 51.68$^\dagger$ | **84.86** | 48.74 |
| Mean ranking | 3 | 2.5$^\dagger$ | 3 | 3.5 | 2.33$^\dagger$ | **1.5** | **1.17** | 2.5$^\dagger$ |

Table 8: Temporal generalization of language models in different training scenarios on the MisInfoText dataset. Best accuracies and F1 are in bold, and second-best results are highlighted by the † symbol.

| Model | R Acc | R F1 | T Acc | T F1 | F Acc | F F1 | TG Acc | TG F1 |
|---|---|---|---|---|---|---|---|---|
| BERT base | **77.14** | 53.63 | 76.99† | 53.23 | 74.00 | **55.50** | 75.97 | 55.30† |
| BERT large | 77.09† | 50.72 | **77.39** | 50.90 | 71.72 | 53.87† | 73.39 | **54.17** |
| DeBERTa base | 77.24† | 51.05 | **77.34** | 50.87 | 73.65 | 54.40† | 74.15 | **55.30** |
| DeBERTa large | 77.29† | 50.79 | **77.39** | 50.90 | 73.50 | 54.48† | 75.16 | **56.60** |
| EuroBERT-210M | **77.34** | 51.39 | 73.39 | 51.76 | 72.64 | **55.47** | 75.92† | 54.49† |
| EuroBERT-610M | 76.58† | 50.09 | **76.73** | 50.11 | 74.30 | **56.28** | 75.37 | 54.05† |
| Mean ranking | 1.66† | 3.66 | **1.5** | 3.33 | 4 | **1.5** | 2.83 | **1.5** |

Table 9: Temporal generalization of language models in different training scenarios on the PHEME dataset. Best accuracies and F1 are in bold, and second-best results are highlighted by the † symbol.

| Model | R Acc | R F1 | T Acc | T F1 | F Acc | F F1 | TG Acc | TG F1 |
|---|---|---|---|---|---|---|---|---|
| BERT base | 53.33 | 34.78 | **56.67** | 36.17 | **56.67** | **42.22** | 53.33 | 40.34† |
| BERT large | **60.00** | **44.10** | 56.67† | 42.22† | 56.67† | 36.17 | 56.67† | 36.17 |
| DeBERTa base | **56.67** | **36.17** | **56.67** | **36.17** | 53.33 | 34.78 | **56.67** | **36.17** |
| DeBERTa large | 53.33 | 34.78 | 56.67† | 36.17 | **60.00** | **48.86** | 56.67† | 46.65† |
| EuroBERT-210M | **56.67** | 36.17† | **56.67** | 36.17† | **56.67** | **42.22** | 53.33 | 34.78 |
| EuroBERT-610M | **56.67** | **36.17** | **56.67** | **36.17** | 50.00 | 33.33 | 50.00 | 33.33 |
| Mean ranking | 1.83† | 2.16† | **1.33** | **2** | 2 | 2.16† | 2.5 | 2.5 |

Table 10: Temporal generalization of language models in different training scenarios on the PolitiFact dataset. Best accuracies and F1 are in bold, and second-best results are highlighted by the † symbol.

| Model | R Acc | R F1 | T Acc | T F1 | F Acc | F F1 | TG Acc | TG F1 |
|---|---|---|---|---|---|---|---|---|
| BERT base | 84.62 | 58.21 | **92.31** | **67.95** | 87.18 | 60.76 | 89.74† | 63.89† |
| BERT large | 89.74† | 63.89† | 84.62 | 58.21 | 87.18 | 60.76 | **94.87** | **73.65** |
| DeBERTa base | 92.31† | 67.95† | 92.31† | 67.95† | 92.31† | 67.95† | **94.87** | **73.65** |
| DeBERTa large | **94.87** | **73.65** | 89.74 | 63.89 | **94.87** | **73.65** | 89.74 | 63.89 |
| EuroBERT-210M | 72.94 | 56.10 | 84.62 | 58.21† | **94.87** | **73.65** | **94.87** | 48.68 |
| EuroBERT-610M | **97.44** | 49.35 | 94.87† | 48.68 | 94.87† | **73.65** | 94.87† | **73.65** |
| Mean ranking | 2.33 | 2.5 | 2.5 | 2.66 | 2† | **1.5** | 1.83 | 2† |

Table 11: Temporal generalization of language models in different training scenarios on the Proppy dataset. Best accuracies and F1 are in bold, and second-best results are highlighted by the † symbol.

| Model | R Acc | R F1 | T Acc | T F1 | F Acc | F F1 | TG Acc | TG F1 |
|---|---|---|---|---|---|---|---|---|
| BERT base | 16.76 | 16.74 | 13.45 | 13.24 | 17.54† | 17.54† | **21.83** | **21.74** |
| BERT large | 19.10 | 19.10 | **27.10** | **26.53** | 20.08† | 20.05† | 18.91 | 18.91 |
| DeBERTa base | 54.58† | 47.71† | 31.77 | 30.52 | 53.80 | 47.15 | **54.97** | **47.99** |
| DeBERTa large | **60.62** | **52.04** | 44.44 | 40.38 | 47.76† | 42.80† | 46.39 | 41.81 |
| EuroBERT-210M | **49.51** | **44.07** | 36.65† | 34.45† | 18.91 | 18.91 | 17.15 | 17.14 |
| EuroBERT-610M | 28.65 | 27.89 | 28.07 | 27.38 | 28.85† | 27.99† | **31.97** | **30.60** |
| Mean ranking | **2.33** | **2.16** | 3.16 | 3.16 | **2.33** | 2.33† | **2.33** | 2.33† |

Table 12: Ranking of the different approaches on the different datasets. The mean ranking over tested models is used and the best performing model for each dataset (in terms of F1 score) are reported.

| | Random | Temporal | Fusion | TempoGen | Best approach |
|---|---|---|---|---|---|
| GossipCop | $2^{\dagger}$ | 4 | **1** | 3 | Fusion - BERT large |
| ISOT | **1** | $2^{\dagger}$ | $2^{\dagger}$ | 4 | Random - DeBERTa base |
| MediaEval | $2^{\dagger}$ | 4 | **1** | $2^{\dagger}$ | Random - EuroBERT-610M |
| MisInfoText | 4 | 3 | **1** | **1** | TempoGen - DeBERTa large |
| PHEME | $2^{\dagger}$ | **1** | $2^{\dagger}$ | 4 | Fusion - DeBERTa large |
| PolitiFact | 3 | 4 | **1** | $2^{\dagger}$ | Multiple including Random, Fusion and TempoGen |
| Proppy | **1** | 4 | $2^{\dagger}$ | $2^{\dagger}$ | Random - DeBERTa large |
| FC30 | 3 | 4 | **1** | $2^{\dagger}$ | TempoGen - DeBERTa large |
| Mean ranking | $2.25^{\dagger}$ | 3.25 | **1.375** | 2.5 | |

# D  MODELS PERFORMANCE OVER TIME

The performance curves for all tested models on FC30 over time are given in the following Figure 5. The oberved performance dip for all models in the 2005-2010 time window may be due to the fast development of the internet at this period, making earlier information *historical*, and this time window less documented when compared to the period after 2010.

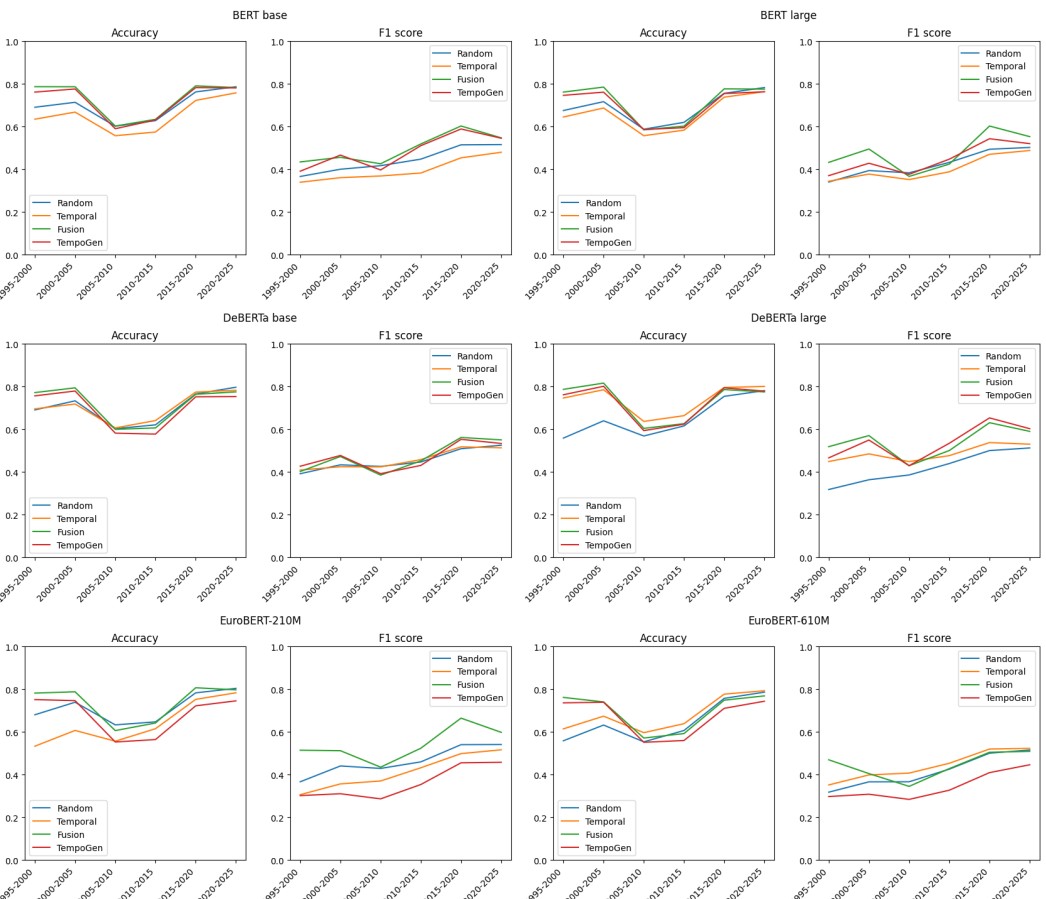

Figure 5: Models' performance over time for different training scenarios.

