# OpenReview forum: "Temporal Misinformation Detection: Simple Ways to Improve Temporal Generalization and Better Evaluate Language Models"
_ICLR.cc/2026/Conference — Submitted to ICLR 2026_

### Official Review · Reviewer_gfbq · 2025-10-30

**Soundness:** 2
**Presentation:** 2
**Contribution:** 2
**Rating:** 4
**Confidence:** 4

**Summary:**

The paper argues for the importance of taking temporal drift into account when studying (and especially evaluating) methods for detecting misinformation.  The paper includes analysis, specifically showing how taking the timestamps of benchmark data into account (train on earlier, test on later examples) leads to reductions in accuracy estimates.  The paper describes a new dataset designed to train on earlier and test on later instances, and introduces new methods based on BERT-style encoder models finetuned on the new dataset, showing how an ensemble of representations trained on temporally organized data and those trained on randomly split data can lead to a performance boost.

More details:

This paper highlights that current misinformation detection models are not evaluated on their performance in real world settings where they may be used to detect misinformation about data that occurred after the model information cut-off date. The authors define temporal generalization as evaluation under three constraints: the data must be temporally ordered, the test data must be constrained such that the earliest item in the test data is later than the latest item in the train and validation splits, and the earliest item in the test data must also be later than the pretraining date of the evaluated model. The authors evaluate current misinformation benchmarks on their ability to correctly find misinformation under these practical constraints. They find that when using DeBERTaV3-Large, most misinformation detection datasets had a lower accuracy and f1-score in this realistic setting. The authors also propose a metric (LabDrift) for measuring whether a dataset is temporally biased. They create a measure of how well a model could do if they were to classify everything prior to a certain date as one label and everything after as the other label. The ratio between the real ordering of the data time stamps, and a perfectly separated ordering provides the LabDrift score. They find that most datasets have some level of temporal bias in the datasets, and provide a new dataset that has the lowest amount of temporal bias. This new dataset is covered by the three constraints that the authors defined, and covers 30 years of data. The authors further propose Fusion and TempoGen as approaches to improve temporal generalization. Both methods combine models trained on a random train/validation split and a temporally ordered train/validation split, where all validation data is later than all train data. While both approaches require twice the train compute, as two different models need to be trained, fusion uses both models at inference time as well, while TempoGen only requires the random model at inference time. Using an F1-score to compare, Fusion is the best approach across several different models, with TempoGen coming in second but with a lower inference cost.

**Strengths:**

The claim that temporal ordering of a dataset could lead to better real world outcomes is strong and sensible tests are conducted to support it

The paper tackles an important problem and draws attention to the temporal issue in dataset construction.

The proposed approaches to mitigating temporal bias include two options, one of which increases training cost (but not inference time cost) and the other which is overall more costly but has stronger performance.  The reviewer appreciates the consideration of computational costs, even though that’s not a major focus of the paper.

Experiments show across three encoder models (variants of BERT) that using embeddings learned on a fusion of random split between training and validation, and a temporal split between training and validation, slightly outperforms using only one split or the other.

Each of the given constraints that ensure temporal correctness is well defined and builds off of the other constraints

**Weaknesses:**

Main technical weaknesses

It is not clear to the reviewer (until reading section 5) what the practical difference is between Constraint TO and Constraint TC. Suggest to clarify more when first introduced

The setup in 3.1, where training/validation/test splits are designed around timestamps, is presented as novel (“it has not been explored in NLP yet” - line 161).  But this has been widely done in papers going back more than a decade at least, especially where the task requires some kind of forecasting.  Closer to this paper’s subject, see Luu et al., 2022, cited in this paper’s introduction.

Section 3.2 experiments don’t give much detail at all, or motivation, for the choice of model.  Was there model selection using the validation set?  How many random seeds for the classical evaluation protocol?  The one dataset where the trend is reversed – MisInfoText – is not discussed at all, though the scores are much higher in the “realistic temporal” evaluation.  Also, why do FC30 numbers go up for F1 and down for accuracy, with the new evaluation setup compared to the old one?

In section 3, the phrase over evaluated in this context does not make sense. I understand that the meaning is that the model performance on classical evaluation metrics is better than in the practical, temporally ordered setting, but "evaluated" is too overloaded a term here.

The labdrift metric in section 4 is designed to measure the extent to which the timestamps in the data predict the labels.  While the idea is reasonable, one could imagine simpler approaches, based on correlations (point-biserial correlation or Eta; or something nonparametric like Mann-Whitney or Kruskal-Wallis), or based on information theory (e.g., what is the entropy of the label distribution, conditional on time?), or just taking tools that already exist in the literature like the temporal degradation quantity from the Luu et al. paper cited in the introduction.  Regardless, Table 2 might be more valuable if it included an estimate of the distribution over labdrift scores for variations of the datasets in which labels were scrambled across items.  This would give the reader a sense of how much “bias” we would see if the datasets were constructed specifically to avoid it.  Some amount is surely expected due to chance (as we see with the nonzero value for FC 30).

The new dataset sounds exciting, but not much information is given about it.  The relationship to other datasets like the Politifact data that’s compared to is not discussed.  What circumstances led to this dataset’s availability now? What steps have the authors taken to ensure that it doesn’t contaminate the training data for new models?  And why couldn’t the existing benchmarks simply be repurposed under the constraints proposed in section 3.1?  The paper doesn’t say what the labels are or how they compare with other datasets for misinformation research, or other datasets that have been designed to challenge models’ temporal generalization.

Statistical significance tests are used in some places but not others; this inconsistency is not explained

The accuracy metric in the tables is not discussed in the main text, and accuracy and F1 scores often don’t point in the same direction in this paper’s results.  This goes completely unremarked and makes me concerned that all of the findings are metric-dependent

The paper doesn’t really engage with or compare to recent temporal alignment methods like “set the clock” in Zhao et al. (“Set the Clock: Temporal Alignment of Pretrained Language Models,” ACL 2024) or the work of Zhang and Choi (“Mitigating Temporal Misalignment by Discarding Outdated Facts, EMNLP 2023) or “mind the gap” (Lazaridou et al., NeurIPS 2021).  It’s hard to know whether the results here represent the state of the art in mitigating temporal misalignment.

The results are also only evaluated on encoder only models, saying that they are “proficient baselines” but do not describe further what that means, or if the increase in abilities of LLMs since the publication of the citation for that claim (2021) has rendered it null, and does not address the obvious question of whether the same findings would hold up with today's more widely used decoder-only models
The paper would be stronger with a tighter connection between the methods and real world practice

Suggestions to improve writing

There is a mix between active and passive voice in the paper that leads to an occasional implied level of distance from the results (“several observations can be made” vs “we make several observations”)

There is a missing “W” in Q2: “...bias in labels. e propose a…”
the introduction and related works sections are written very informally and have some awkward transitions between tasks (one example being “Many datasets exist (D’Ulizia et al., 2021), allowing for the training of models to detect different aspects of misinformation” what are the datasets?)

In section 1 when defining the research questions, the phrase “temporal test set” is used without definition. Though temporality is defined, and it is clear that this temporal test set is different from a random split, it is not clear what it is.

**Questions:**

When you concatenate the embeddings and train the two-layer perceptron, do you know how similar the performance of the model is on each item when compared to the individual models? I’d be interested in a per-question evaluation of how often the fusion is the same as the random vs the same as the temporal models

Table 1 reports many statistical significance tests (about 16 by my counting).  What corrections were used to account for multiple tests?

---

### Official Review · Reviewer_QBAT · 2025-10-31

**Soundness:** 2
**Presentation:** 1
**Contribution:** 1
**Rating:** 2
**Confidence:** 4

**Summary:**

The paper argues that random-split evaluation inflates misinformation-detection results and that models should be tested under temporal generalization: train on past, validate on a temporally earlier slice, and test strictly on later data (Constraints TO, TC, PT). It introduces LabDrift, a label-drift metric based only on temporally ordered labels; releases FC30, a 36,619-claim fact-checking dataset spanning 1995–2025 with labels aggregated to three classes; and proposes two simple training strategies—Fusion (concat embeddings from random- and temporal-trained encoders plus an MLP) and TempoGen (train the MLP on both embeddings but use only temporal embeddings at inference)—to improve temporal robustness.

**Strengths:**

1. The TO/TC/PT constraints crisply capture leakage risks from both fine-tuning and pre-training.

2. The paper is easy to follow

**Weaknesses:**

1. The paper studies an existing problem. The paper does not discuss the difference between existing works such as [1][2].

[1] Jiang B, Tan Z, Nirmal A, et al. Disinformation detection: An evolving challenge in the age of llms[C]//Proceedings of the 2024 siam international conference on data mining (sdm). Society for Industrial and Applied Mathematics, 2024: 427-435.

[2] Jiang B, Zhao C, Tan Z, et al. Catching chameleons: Detecting evolving disinformation generated using large language models[C]//2024 IEEE 6th International Conference on Cognitive Machine Intelligence (CogMI). IEEE, 2024: 197-206.


2. Core experiments use encoders (BERT/DeBERTa/EuroBERT). The case for “LLMs degrade under temporal shift” would be stronger with decoder-only baselines or recent instruction models.

3. Fixing the test to the last 10% may be brittle for smaller corpora/classes (the paper notes PolitiFact’s tiny temporal test). Consider rolling-window or multiple cut points.

4. FC30 pulls from Snopes/PolitiFact; generalization to social-platform posts, multimodal content, or non-English settings is untested.

5. On FC30, some improvements are modest and model-dependent (Table 3). Error analyses (topic drift, named entities, lexical shift) would clarify where methods help or fail.

**Questions:**

1. Can u discuss in more detail the related/existing works?

2. Can you report variance across multiple temporal cut points (e.g., last 5/10/20%) and rolling-origin evaluation to reduce dependence on a single split?

3. Do results transfer to decoder-only LLMs (frozen features + linear probe) and to multimodal detectors?

4. On FC30, can you provide topic/entity drift analyses and qualitative failure cases for each approach?

---

> ### Author Response · Authors · 2025-11-17
>
> Dear reviewer QBAT,
>
> Thank you for your review.
> We want to clarify some points and answer some remarks:
>  - Remark 1 and Question 1 and 3: The scope of the study focuses on embeddings produced by encoder models for text classification. While other work on decoder models is interesting and kept in mind for future work, we focus on temporal generalization in classification tasks, which are not yet reliably solved by decoder models. Moreover, the two proposed papers differ in their approach to temporal generalization; the one we consider is concerned with the evolution of real-world events in general, rather than the evolution of the methods of producing misinformation, which may be observable in some datasets but not in FC30.
>  - Remark 3 and question 2 on the test data: Thank you for the suggestion. We will keep a note of it for future research, as we do not have enough space in the current format for these additional experiments. The choice of 10\% of test data in the Temporal settings has been made to mirror the classical random split proportions. While other proportions can be tested, we chose to keep the one that is closest to a random split.
>  - Remark 4 on generalization to other data types: While FC30 is more focused on fact-checking, we tested other data types (press articles, claims, and tweets, depending on the datasets), but did not specify them beyond the dataset name. We will add the data types and tasks in the corresponding tables in the Appendix.
>  - Question 4 on providing topic/entity drift analyses and qualitative failure cases for each approach: thank you for the suggestion, we indeed think this would be helpful for further improving our approach, but unfortunately, we do not have the space to include it in the main paper to leave room for the other contributions.
>
> If some of your questions are not fully answered or if you require further clarification, please respond to this comment.

---

> > ### Comment · Reviewer_QBAT · 2025-11-24
> >
> > It seems to me that the two papers also focus on embeddings produced by encoder models for text classification and temporal generalization.

---

### Official Review · Reviewer_kAEq · 2025-10-31

**Soundness:** 2
**Presentation:** 3
**Contribution:** 2
**Rating:** 2
**Confidence:** 4

**Summary:**

This paper addresses the problem of temporal generalization in misinformation detection models. The authors argue that standard evaluation protocols, which rely on random data splits, tend to overestimate model performance. This is because models may learn event-specific representations rather than generalizable patterns, a critical flaw in a real-world setting where new misinformation constantly emerges.

**Strengths:**

1. The paper's clearest strength is its empirical diagnosis of the overestimation problem. Table 1 provides stark, quantitative evidence of how much traditional benchmarks inflate performance, with F1 scores dropping by as much as 40-50 points (e.g., MediaEval, Proppy).

2. The paper introduces FC30, a new dataset with a 30-year span. Its most important feature is the large amount of data from 2020-2025, which finally allows for proper evaluation of models under the "Pre-Training" (PT) constraint.

**Weaknesses:**

1. The paper's main conclusion about the effectiveness of Fusion and TempoGen is based on FC30. However, the appendix (Table 12) shows these methods are not a general solution. On ISOT, MediaEval, and Proppy, the 'Random' baseline performs best. This lack of generalization suggests the methods may be tuned to the specific properties of FC30 and are not robust. This is the most significant weakness.

2. The central thesis that "models must generalize to future data" is a well-known form of distribution shift. While applying it to misinformation is valid, the paper overstates its novelty. Many recent works on LLM evaluation, robustness, and "out-of-distribution" (OOD) detection have highlighted this exact challenge. The paper fails to differentiate its contribution from this large body of existing work.

3. The results in Table 3 show that the 'Temporal' (T) split baseline consistently performs worse than the 'Random' (R) split baseline. This is counter-intuitive, as one would expect the 'T' split to be better aligned with the temporal test set. The paper does not investigate why this happens. Is the model learning spurious temporal cues from the validation set? Without this analysis, the proposed solution (which just combines R and T) feels like an ad-hoc fix rather than a principled one.

4. Furthermore, I personally believe that the introduction of such benchmarks necessitates measures to prevent benchmark data contamination. Otherwise, such datasets will become obsolete after the next large-scale pre-training.

**Questions:**

1. The most critical issue is the discrepancy between your main results (Table 3) and your appendix results (Table 12). Why do Fusion and TempoGen fail to generalize, and in some cases, perform worse than the 'Random' baseline on datasets like ISOT, MediaEval, and Proppy? Does this not invalidate the claim that they are effective methods for improving temporal generalization?

2. Could you provide more analysis on why the 'Temporal' split baseline (T) performs so poorly, even worse than the 'Random' (R) baseline? This seems to be a key finding. Does this imply that naively training on "the most recent" data is actually harmful for generalization, perhaps due to overfitting on temporally-specific artifacts in the validation set?

3. Given that the problem of temporal distribution shift is well-established, could you clarify what you see as the primary conceptual contribution for the community, beyond the (valuable) dataset and metric? The proposed methods seem to be straightforward ensembles rather than a new learning paradigm.

**Details Of Ethics Concerns:**

The authors introduced a new dataset, FC30, in their paper. This dataset was created by collecting 36,619 claims from the PolitiFact and Snopes websites. The authors explicitly stated that they would publicly release this FC30 dataset (line 316).

The primary concern lies in the potential violation of the terms of use and copyright policies of the source websites (PolitiFact). The paper fails to address whether its data collection (web scraping) and redistribution activities comply with these websites' policies. The authors provide no evidence demonstrating they obtained lawful authorization to scrape and redistribute this copyrighted content, raising serious legal compliance issues.

For example, on PolitiFact webpage (https://www.politifact.com/copyright/), they mention:
- "Storing or archiving any significant portion of the content or creating a database using the content is prohibited."
- "Any other activity or use of this site that is a violation of United States intellectual property laws or international treaties, including but not limited to copying other than on an isolated basis for personal use; modifying the content; and publishing, transmitting, creating derivative works from, transferring, selling or displaying the content of the site, is prohibited."

---

> ### Author Response · Authors · 2025-11-17
>
> Dear reviewer kAEq,
>
> Thank you for your review. We would like to respond to some of your comments:
>  - Question 1 on generalization: The main explanation we don't provide in the main article, but should have, is that the proposed Fusion and TempoGen methods aim at improving the temporal generalization of tasks for which features are inherently time-dependent. ISOT is a relatively biased dataset on sources, which means that the most relevant features are not time-dependent. The same goes for MediaEval (credibility estimation) and Proppy (propaganda). While the proposed approach is tested on specific subtasks of misinformation, it is only applied to those that are mostly time-dependent (fake news and rumour detection). The claim is not invalidated, but it should be specified to make the time dependency more prominent.
>  - Question 2 on the performance of the temporal split alone: By selecting features temporally, we indeed overfit on temporally-specific artifacts from the validation time period. This can be seen as similar to the bias-variance tradeoff: selecting features temporally makes them closer to the test distribution in terms of time period (low variance). Still, they are not always aligned (high bias), which makes the use of temporally independent variables simultaneously required.
>  - Question 3 on the contribution: while the problem has been stated before, it has been only explored superficially for NLP tasks, as learning temporal language representation is a lot more difficult than learning distribution shifts and language representation separately. While relatively straightforward, the proposed approaches quantitatively demonstrate the limitations of using random features and suggest that temporal features are also insufficient by themselves, but that combining them enables performance improvements for temporally sensitive tasks. Generally, while the temporal distribution shift is a well-established problem, very little research takes it into account, and reported scores are almost never aligned with what would be observed in an applicative setting. We want to emphasize this further by measuring performance drops and showing that not all tasks are affected equally.
>  - Weakness 2: The main difference with robustness works and OOD generalization is that, while similar, these problems are different, and previously proposed approaches only worsen the temporal generalization performance, as shown in Su et al. (2022), mentioned in the related work section line 98 on the failure of domain adaptation for temporal generalization.
>  - Weakness 4: This is one point we tried to make in the paper, but did not insist upon. Constraint (PT) is important for a fair evaluation of performance. However, due to the public nature of the mentioned facts, contamination is inevitable for models trained in the future, which is why the benchmark will need to be updated regularly to maintain an accurate evaluation of the models.
>  - On the Ethics flag: We would like to correct some of our claims on how the dataset would be shared. First of all, we did not collect a significant portion of the Snopes or PolitiFact databases, retaining only the external claims, labels, and publication dates. To ensure compliance with all relevant legislation regarding data sharing, we will reduce the dataset and provide only the URLs of the fact-checks, as well as the labels we consider for training ("true", "false", and "mixed"), which are not extracted from the websites. We would like to clarify that we will not share any copyrighted materials produced by Snopes and PolitiFact, and that no content other than the claims themselves was used or stored.
>
> If some of your questions remain unanswered or if you require further clarification, please respond to this comment.

---

### Meta-Review · Area_Chair_yhWn · 2026-01-05

**Summary:**

The paper presents an in-depth study on temporal generalization of fact checking datasets. They first analyze how taking timestamps of benchmark into consideration (test data to be later than train/val data) significantly reduces the model performances across the board. The author later proposes methods for fine-tuning encoder models on temporally organized data, showing this can improve the temporal generalization.

The reviewers have provided rich feedback. Specifically, reviewer asked for comparison with related work (both in terms of framing of this work’s novelty, discussion of different methods, or the presented data) (pointed by all three reviewers gfbg, kAEq, QBAT). I do not think author's response towards this concerns were very satisfactory (simply asserting that there exist prior work for showing the issues but not mitigation efforts, while Reviewer gfbg concretely provide some prior work in mitigation efforts). I also think the distinction authors are making (temporal generalization in encoder model vs. decoder model) is very in-depth, as neither mitigation strategy nor the nature of the problem heavily depends on this architectural difference. In its current form, it is hard to recommend paper acceptance.

**Reviewer Concerns:**

See meta review.

**Reviewer Scores:**

I do not think reviewers would have changed the scores, given the extent of author rebuttal (no new results / analysis presented). For Reviewer gfbq, who provided in-depth review, the authors did not provide response.

---

### Decision · Program_Chairs · 2026-01-26

Reject